# Surrounding Semi-Natural Vegetation as a Source of Aphidophagous Syrphids (Diptera, Syrphidae) for Aphid Control in Apple Orchards

Elżbieta Wojciechowicz-Żytko [1,*] and Edyta Wilk [2]

1 Department of Botany, Physiology and Plant Protection, Agricultural University, al. 29-Listopada 54, 31-425 Kraków, Poland
2 Voivodeship Inspectorate of Plant Health and Seed Inspection, ul. Langiewicza, 2835-101 Rzeszów, Poland; edyta.w85@interia.pl
* Correspondence: elzbieta.wojciechowicz-zytko@urk.edu.pl

**Abstract:** The influence of different semi-natural surroundings adjacent to apple orchards on the occurrence of predatory syrphids and biological control of *Aphis pomi* Deg. and *Dysaphis plantaginea* Pass. was compared. Two methods of catching hoverflies were used: yellow traps to collect the adults and hand picking to get the larvae from aphid colonies. A total of 1029 Syrphidae (26 species belonging to 14 genera) of subfamily Syrphinae were collected in Moericke traps from apple orchards and their boundaries. At all sites, a much greater number of hoverflies was collected in the surroundings (638 specimens) than in the orchards (391 specimens). In apple orchards, 134 syrphids belonging to 10 species were reared from *A. pomi* and *D. plantaginea* colonies. In both cases, the dominants were *Episyrphus balteatus* (Deg.) *Eupeodes corollae* (Fabr.), *Syrphus vitripennis* Meig. and *S. scripta* (L.), suggesting that hoverflies are attracted by plants flowering in semi-natural habitats in the vicinity of the orchard, and they then migrate to the orchard and reduce the aphid colonies. The results confirmed the positive influence of natural surroundings on the conservation of aphid predators.

**Keywords:** predators; semi-natural habitats; biological control; aphids; apple orchards

## 1. Introduction

The challenge facing agriculture in the coming years is to produce a good yield of healthy food using improved environmentally friendly practices [1,2]. The main tasks of integrated protection include increasing the presence of suitable places for the development of parasites and predators and thus enhance pest control. Habitat management aims to meet both agronomic and ecological goals by regulating insect pests, including by intensifying the predator's natural impact and by preserving and promoting biodiversity [3–6].

In apple orchards, a high level of pesticide use is required to control the pest burden. An increasing number of studies show the adverse side effects of pesticides on beneficial insects, which include a shorter life and lower fecundity [4,7–9]. Therefore, a very important issue is the limitation of the use of chemical agents and the use of alternative (mechanical, physical, or agrotechnical) methods for controlling pests [10–12].

Aphids are an important biotic stress factor negatively affecting the quality and quantity of the fruit crop [13]. Some aphid species on apple trees (rosy apple aphid *Dysaphis plantaginea* Pass and green apple aphid *Aphis pomi* Deg. (Hemiptera, Aphididae)) are the most dangerous pests, causing the deformations of leaves and fruits, twisting of shoots and chlorosis. They also weaken trees and make them less frost-resistant [13].

Syrphids (hoverflies) (Diptera, Syrphidae) play an important role as pollinators (adults) and predators (larvae). The flies need the pollen and nectar of flowers for reproduction so they pollinate the plants while their predatory larvae (subfamily Syrphinae) reduce the

population of aphids on crop plants [6,14–17]. Highly reproductive hoverfly females find even small colonies of aphids, and the developing larvae quickly destroy their prey [18].

Monoculture crops do not provide a source of food for predators and parasites. An important factor increasing the presence of these insects is the diversification of the vegetation around fields and orchards to provide a habitat for natural enemies. Refugia in the neighborhood of crops can improve the biodiversity of beneficial insects, which, during the vegetation season, pollinate crop plants and reduce pest populations [19–23].

The survival of beneficial insects in the agricultural environment depends on the presence of semi-natural habitats. The biodiversity of beneficial insects can be modeled by the features of the habitat. Refugia increase the number of beneficial insects, being a place of sheltering or overwintering for them as well as a source of alternative prey [5,14,24–27].

In practice, it is possible to influence the agricultural landscape by creating the appropriate types of habitats attractive to adult Syrphidae in the vicinity of orchards, and therefore increase the population of these predators [6,22,25,28,29]. On the other hand, the existing semi-natural habitats in the vicinity of orchards should be protected and improved as a source of biodiversity of beneficial insects. Knowledge of the contribution of each environment is necessary to develop control strategies.

Semi-natural habitats in the surroundings of crops have been shown to be reservoirs of biodiversity [30–34], evidenced by the penetration of beneficial organisms from these natural areas into cultivated habitats [34,35]. Happea et al. [36] presented the influence of landscape composition on the abundance and community composition of predatory insects in apple orchards in three European countries and stated that semi-natural habitats enhanced the presence of beneficial insects and their contribution to biological control in fruit crops. Daelemans et al. [37] tested the effects of landscape composition on natural enemy communities and the pest control services they provide in apple orchards. Authors suggested that habitat conservation can enhance natural enemies and stimulate pest control.

However, there has been little research conducted on the influence of natural orchard margins on the occurrence and the species composition of hoverflies, while their impact on *D. plantaginea* and *A. pomi* pests was rarely investigated. Therefore, the following hypotheses were tested: (1) the composition of predatory syrphids differs between orchards and orchards surroundings; (2) predatory syrphids benefit from enhanced landscape diversity; (3) the abundance of hoverflies is higher in orchards with many plant species in the surroundings; and (4) semi-natural surroundings increase the number of hoverflies feeding in aphid colonies on apple trees.

The aims of this work were to determine the species composition of predatory syrphids occurring in yellow traps in orchards, surroundings and in aphids colonies, and to determine the influence of the surrounding vegetation of orchards on the appearance of zoophagous species of hoverflies reducing aphid populations on apple trees.

## 2. Materials and Methods

### 2.1. Research Sites

The current study was conducted in the years 2011–2013 in the fruit tree-growing area in the south-eastern part of Poland near the town of Przemyśl (49.82° N, 22.79° E), in three apple orchards with integrated pest management (IPM) with 'Szampion', 'Elise' and 'Elstar' cultivars as well as in their semi-natural surroundings (spontaneous vegetation including herbaceous plants, trees, and shrubs). The distance between the IPM orchards was 1–1.5 km.

Every 'site' consisted of the orchard and its surroundings.

Site 1—The IPM orchard of 9 ha surrounded by woodlands (sour cherry *Cerasus vulgaris* Mill., English walnut *Juglans regia* L., Norway spruce *Picea abies* (L.), *cherry Prunus* spp. L., *staghorn sumac Rhus typhina* L.), shrubs (common barberry *Berberis vulgaris* L., common hazel *Corylus avellana* L., common hawthorn *Crataegus monogyna* Jacq., wild privet *Ligustrum vulgare* L., black chokeberry *Photinia melanocarpa* Michx., gooseberry *Ribes uva-crispa* L., dog rose *Rosa canina* L., raspberry *Rubus* L., elderberry *Sambucus nigra* L., common

lilac *Syringa vulgaris* L.) and herbaceous plants (common yarrow *Achillea millefolium* L., ground elder *Aegopodium podagraria* L., absinthe *Artemisia absinthium* L.; shepherd's purse *Capsella bursa-pastoris* L., wild carrot *Daucus carota* L., smallflower galinsoga *Galinsoga parviflora* Cav., cleavers *Galium aparine* L., white nettle *Lamium album* L., wild chamomile *Matricaria discoidea* DC., narrowleaf plantain *Plantago lanceolata* L., meadow buttercup *Ranunculus acris* L., common buckthorn *Rhamnus cathartica* L., sorrel *Rumex acetosa* L., chickweed *Stellaria media* (L.) Vill., European goldenrod *Solidago virgaurea* L., dandelion *Taraxacum officinale* Web., white clover *Trifolium repens* L., stinging nettle *Urtica dioica* L., bird's-eye speedwell *Veronica chamaedrys* L.);

Site 2—IPM orchard of 10 ha, surrounding by a pear orchard (pear *Pyrus* L.) (also with IPM) and herbaceous plants (common yarrow *Achillea millefolium* L., ground elder *Aegopodium podagraria* L., shepherd's purse *Capsella bursa-pastoris* L., wild carrot *Daucus carota* L., smallflower galinsoga *Galinsoga parviflora* Cav., cleavers *Galium aparine* L., white nettle *Lamium album* L., wild chamomile *Matricaria discoidea* DC., narrowleaf plantain *Plantago lanceolata* L., meadow buttercup *Ranunculus acris* L., sorrel, garden *Rumex acetosa* L., common chickweed *Stellaria media* (L.) Vill., European goldenrod *Solidago virgaurea* L., dandelion *Taraxacum officinale* Web., white clover *Trifolium repens* L., stinging nettle *Urtica dioica* L., bird's-eye speedwell *Veronica chamaedrys* L.);

Site3—IPM orchard—8.5 ha, surrounded by a pear orchard (also with IPM), woodlands with a predominance of sour cherry *Cerasus vulgaris* Mill., Norway spruce *Picea abies* (L.), cherry *Prunus* L., *staghorn sumac Rhus typhina* L., common barberry *Berberis vulgaris* L., common hazel *Corylus avellana* L., wild privet *Ligustrum vulgare* L. and herbaceous plants (ground elder *Aegopodium podagraria* L., wild carrot *Daucus carota* L., white nettle *Lamium album* L., narrowleaf plantain *Plantago lanceolata* L., meadow buttercup *Ranunculus acris* L., *European buckthorn Rhamnus cathartica* L; common chickweed *Stellaria media* (L.), dandelion *Taraxacum officinale* Web., white clover *Trifolium repens* L., stinging nettle *Urtica dioica* L., bird's-eye speedwell *Veronica chamaedrys* L.).

The arrangement of the trees in the examined orchards was similar with the spacing of 1.5 m × 3 m and a grass sward between the rows of trees. The width of the orchard surroundings was 7–8 m. The plants growing in the semi-natural habitats were manually identified. During the period of the study, in the orchards with integrated pest management (IPM), the apple trees were protected using the pesticides and fungicides in accordance with the methodology of integrated production (5–7 treatments against diseases and 6–7 against pests).

*2.2. Method of Sampling*

2.2.1. Sampling of Aphids

To assess the presence of green apple aphid A. pomi and rosy apple aphid D. plantaginea, 100 marked shoots per orchard (10 shoots per tree were randomly selected from different parts of the crown in 10 trees, and placed up to 20 m away from the margin) were inspected from the end of April to October (at 10–14-day intervals). In small colonies, all the aphids including alate and apterous were counted; when they were very numerous, their number was estimated. On this basis, the population dynamics of both species was established.

2.2.2. Sampling of Syrphidae

Two sampling methods were used:

Yellow Traps

Moericke traps were used to collect syrphid adults. Each trap consisted of a yellow dish containing water with glycol [38]. Traps were hung on branches 1.5–2 m above the ground level; 10 traps were placed randomly in the middle in each orchard and 10 in its boundaries, every 10 m. To avoid the impact of marginal effects, the traps were placed about 10–15 m from the edge of the orchard or its surroundings. The traps were checked

from the end of April to the end of September at 10–14-day intervals. One sample consisted of hoverflies collected in a dish within 10–14 days.

Hand Picking

In order to determine the species composition of Syrphidae occurring in Aphis pomi Deg. and Dysaphis plantaginea Pass colonies, larvae and pupae were collected during 30 min from aphid colonies (from the same 10 trees as aphids but from unmarked shoots) from April to July at 10–14-day intervals. The syrphids collected in one day constituted one sample. Larvae were next reared separately in Petri dishes lined with filter paper under laboratory conditions (temperature 23 °C and relative air humidity of 70%), fed daily with aphids until their emergence.

Syrphid adults collected from yellow traps and those from aphid colonies were identified to the species level in the laboratory under the microscope taking into account the specific characteristics of each species, based on the keys of van Veen [39] and Rotheray [40]; the terminology used was according to Soszyński [41].

### 2.3. Statistical Analysis

Species composition, dominance structure, frequency, species richness and similarity of syrphid associations were established. Five dominance classes were identified: >10% eudominants, 5.1–10% dominants, 2.1–5% subdominants, 1.1–2% recedents, and <1% subrecedents were adopted [42].

Species richness was calculated based on the formula:

$$S = s - 1/\log N$$

S—species richness, s—total number of different species, and N—total number of individuals.

The Jaccard classical index [43] was used to calculate the similarity of hoverfly associations.

$$J_{class} = A/A + B + C \tag{1}$$

$J_{class}$—Jaccard similarity index, A—number of shared species, B—number of species unique to the first assemblage, and C—number of species unique to the second assemblage.

In order to assess the relationship between the syrphids (number of specimens as well as the number of species) found in the different habitats, correlations were run between the groups of hoverflies: syrphid larvae captured in aphids colonies, and adult hoverflies collected into yellow traps in orchards and in surroundings. In this case, global analysis was used. Data were analyzed using Pearson correlation coefficients (tests on normal distribution K-S). The differences between the occurrence of the most abundant syrphid species in different habitats were calculated using one-way ANOVA statistics and the Tukey's Post Hoc test. A significance level of <0.05 was considered for all analysis. Data were analyzed using the statistical software package Statistica (Version 13, Kraków, Poland)).

### 3. Results

*3.1. Occurrence of D. plantaginea and A. pomi on Apple Trees*

Differences in the infestation of apple trees by aphids were registered in different sites of observation. The most aphids in all the years of observations occurred in the orchard on site 1; the least numerous colonies were noted in the orchard on site 3. The most common species that infested apple trees in this region in all the years of observations was *A. pomi* (Figures 1–3).

The first small aphid colonies (*D. plantaginea* and *A. pomi*) were noted in the end of April/beginning of May on all sites. During the growing season, the number of aphids increased. Their population rapidly grew in number and reached the maximum in mid-July (*A. pomi*) and in the beginning of June (*D. plantaginea*) on all sites. During this time, *D. plantaginea* established colonies from 11 (site 3) to 17 (site 1) specimens/shoot whereas

*A. pomi* from 33 (site 3) to 47 (site 1) (Figures 1–3). Aphids fed on the top part of young shoots (*A. pomi*) and on the upper side of leaves (*D. plantaginea*). After this time, the number of aphids decreased, which was associated with the activity of predators and aphid migrations.

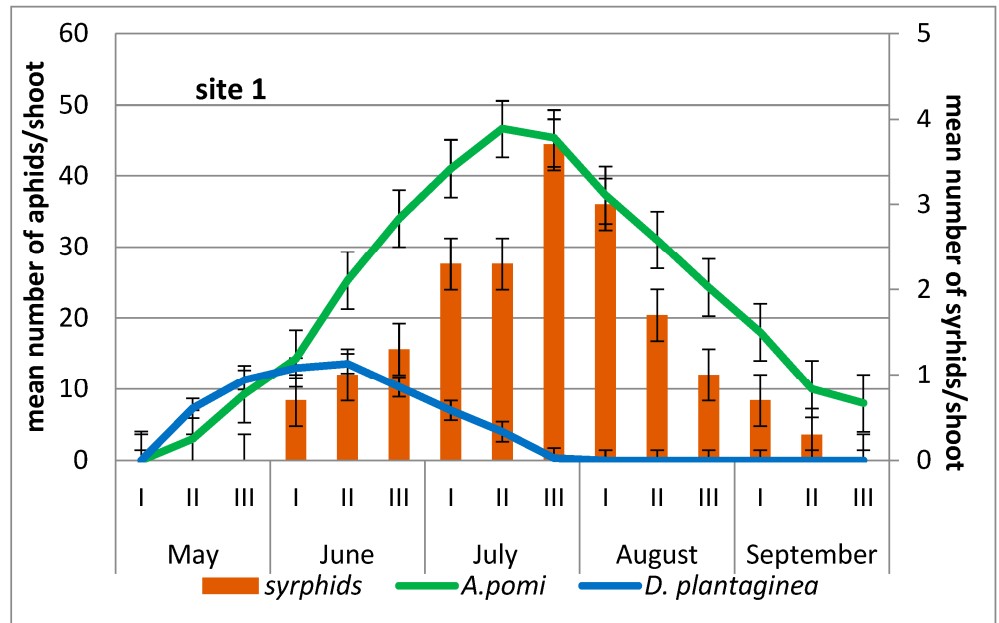

**Figure 1.** Population dynamics of *A. pomi*, *D. plantaginea* and syrphids on site 1 (mean from 3 years).

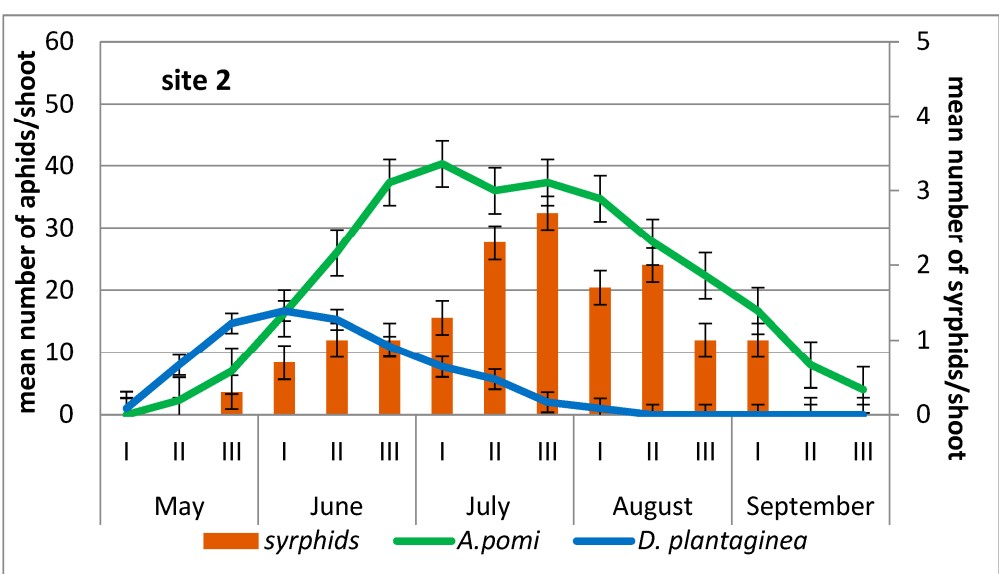

**Figure 2.** Population dynamics of *A. pomi*, *D. plantaginea* and syrphids on site 2 (mean from 3 years).

### 3.2. Syrphids Collected from the Aphid Colonies

Syrphids as a specialized predators occurred in aphid colonies few days later than the pests (Figures 1–3). The first larvae were observed in the beginning of June. Their most numerous abundances were noted from the second to third decade of July (site 3 and sites 1 and 2, respectively). During this time, the predator-to-prey ratio changed from 1:15 (site 3) to 1: 18 (site 2).

The mean number of the Syrphidae (per year) collected from the aphid colonies in the investigated apple orchards (Table 1) and the species composition (Table 2) are shown.

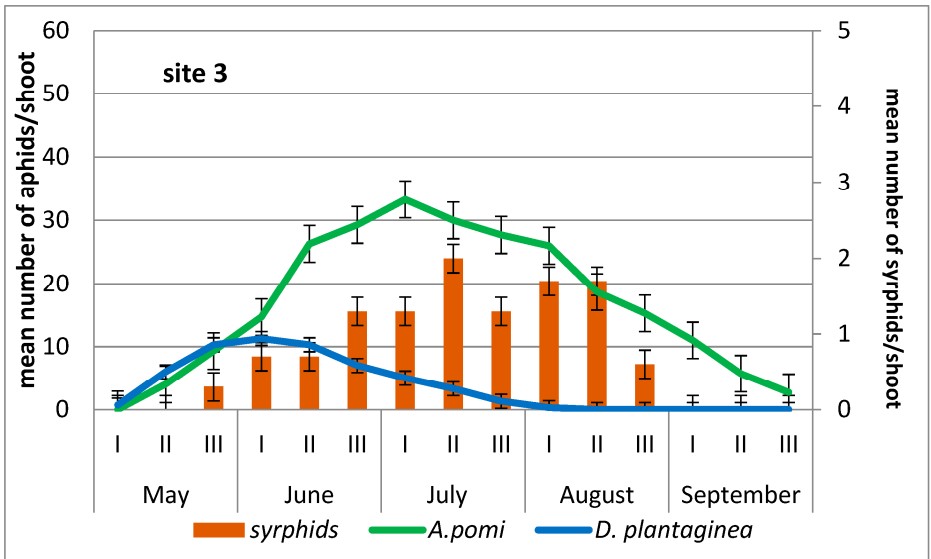

**Figure 3.** Population dynamics of *A. pomi*, *D. plantaginea* and syrphids on site 3 (mean from 3 years).

**Table 1.** Mean (±SE) number of syrphids (species and specimens) collected in yellow traps in orchards and surroundings and found in aphid colonies.

|  | Traps in Orchards | Traps in Surroundings | Aphid Colonies |
|---|---|---|---|
|  | Mean number of specimens | | |
| Site1 | 27.00 ± 16.92 | 54.33 ± 13.38 | 18.00 ± 2.08 |
| Site 2 | 52.00 ± 25.18 | 77.33 ± 52.68 | 15.00 ± 2.52 |
| Site 3 | 50.67 ± 20.70 | 81.33 ± 38.83 | 11.67 ± 2.19 |
|  | Mean number of species | | |
| Site1 | 6.00 ± 2.08 | 9.00 ± 1.73 | 6.33 ± 0.33 |
| Site 2 | 11.00 ± 1.73 | 8.67 ± 1.76 | 5.67 ± 0.88 |
| Site 3 | 4.67 ± 0.33 | 11.33 ± 1.45 | 3.67 ± 0.33 |

**Table 2.** Species composition, domination of predatory syrphids collected from *Aphis pomi* and *D. plantaginea* colonies in different orchards.

| Species | Orchard 1 | | Orchard 2 | | Orchard 3 | | Total | |
|---|---|---|---|---|---|---|---|---|
|  | No | % | No | % | No | % | No | % |
| *Epistrophe eligans* (Harr.) | 3 | 5.5 D | 3 | 6.7 D |  |  | 6 | 4.5 Sd |
| *Epistrophe melanostoma* (Zett.) | 1 | 1.9 R |  |  |  |  | 1 | 0.7 Sr |
| *Episyrphus balteatus* (Deg.) | 24 | 44.4 Eu | 20 | 44.4 Eu | 19 | 54.3 Eu | 63 | 47.0 Eu |
| *Eupeodes corollae* (Fabr.) | 9 | 16.7 Eu | 6 | 13.3 Eu | 9 | 25.7 Eu | 24 | 17.9 Eu |
| *Eupeodes lapponicus* (Zett.) | 1 | 1.9 R |  |  |  |  | 1 | 0.7 Sr |
| *Scaeva pyrastri* (L.) | 1 | 1.9 R | 2 | 4.4 Sd |  |  | 3 | 2.2 Sd |
| *Sphaerophoria scripta* (L.) | 7 | 12.9 Eu | 6 | 13.3 Eu | 3 | 8.6 D | 16 | 11.9 Eu |
| *Syrphus ribesii* (L.) |  |  | 1 | 2.3 R |  |  | 1 | 0.7 Sr |
| *Syrphus torvus* O.-S. | 2 | 3.7 Sd | 2 | 4.4 Sd | 1 | 2.8 Sd | 5 | 3.7 Sd |
| *Syrphus vitripennis* Meig. | 6 | 11.1 Eu | 5 | 11.2 Eu | 3 | 8.6 D | 14 | 10.4 Eu |
| number of specimens | 54 | 100.0 | 45 | 100.0 | 35 | 100.0 | 134 | 100.0 |
| Number of species | 9 |  | 8 |  | 5 |  | 10 |  |

Eudominants (Eu) > 10%, Dominants (D) 5.1–10%, Subdominants (Sd) 2.1–5%, Recedents (R) 1.1–2%, Subrecedents (Sr) < 1%. No—number, %—percentage.

During the observations, 134 individuals belonging to 10 species were reared. The number of the collected species ranged from 5 in the orchard on site 3 (mean 3.67 ± 0.33) to 9 in the orchard on site 1 (mean 6.33 ± 0.33) (Tables 1 and 2).

Throughout the study, the larvae of Syrphidae were the most numerous in the orchard on site 1 (total 54 individuals, mean 18.00 ± 2.08) with diverse vegetation in the surroundings, whereas the least number of individuals and species were noted in the orchard on site 3 (mean 11.67 ± 2.19) (Tables 1 and 2).The most numerous species in all the years of observation and on all sites was *Episyrphus balteatus* (Deg.) whose larvae constituted almost 50% of all collected specimens, along with *Eupeodes corollae* (Fabr.), *Sphaerophoria scripta* (L.) and *Syrphus vitripennis* Meig. They were classified as eudominants.

Some species were scarce—they appeared only in some sites and years—for example *Epistrophe melanostoma* (Zett.) and *Eupeodes lapponicus* (Zett.) occurred only in orchard 1, while *Syrphus ribesii* (L.) only in orchard 2 (Table 2).

Interestingly and unexpectedly, the species richness of hoverflies collected from the aphid colonies in orchard 3 was very low (2.7), although the one observed in the surroundings of this orchard was the highest recorded (8.3) (Table 3).

**Table 3.** Species richness of Syrphidae on different sites.

| | Site 1 | | | Site 2 | | | Site 3 | | |
|---|---|---|---|---|---|---|---|---|---|
| | **o** | **s** | **a.c.** | **o** | **s** | **a.c.** | **o** | **s** | **a.c.** |
| No of species | 13 | 16 | 9 | 16 | 15 | 8 | 8 | 21 | 5 |
| Species richness | 6.3 | 6.8 | 4.7 | 6.8 | 5.8 | 4.1 | 3.2 | 8.3 | 2.7 |

o—orchard, s—surrounding, a.c.—aphid colonies.

### 3.3. Syrphid Collected in Yellow Traps

In this study, a total of 1029 syrphids (26 species belonging to 14 genera) of the subfamily Syrphinae were collected in apple orchards and their boundaries during 3 years of observations (Table 4). At all sites, a much greater number of hoverflies was collected in the surroundings than in the orchards.

The largest amount of syrphid species (21) and individuals (total 244, mean 81.33 ± 38.83) was collected in the surroundings on site 3 where there was a rich vegetation cover, whereas the least number of specimens was noted in orchard 1 (81, mean 27.00 ± 16.92) (Tables 1 and 4).

Amongst all species, *E. balteatus* was the most distributed species (eudominant) especially in the surroundings of site 3. The mean percentage of this species was 43.7% in the apple orchards and 39% in the boundaries. In total, *E. balteatus* accounted for more than 40% of all the caught syrphids (Table 4). Interestingly its percentage contribution in aphid colonies was even higher at 47% (Table 2).

The frequency of *E. balteatus* varied from 25% (in the surroundings on site 2) to 50% (in the surroundings on site 3), i.e., this species was presented in a large number of samples (one-fourth and half, respectively) (Table 4).

Among the other species characterized by a high frequency were *E. corollae* (33.3% in the surroundings in site 2 and 3) *S. scripta* with a frequency of 27.8% in the surroundings of orchard 1, but surprisingly not found in orchard 3, and *S. vitripennis*—16.7% in the surroundings on site 1 and site 3. These three species constituted 27.9% of all collected specimens (Table 4).

Table 4 shows that other species were less numerous and species such *Cheilosia pagana* (Meig.), *Eristalis interrupta* (Poda), *Neoascia podagrica* (Fabr.) and *Xanthandrus comtus* (Harr.) were noted only in the orchards, while *Parasyrphus annulatus* (Zett.) just in the surroundings.

**Table 4.** Species composition, domination and frequency of predatory syrphids collected in the yellow traps in orchards and their surroundings.

| Species | Site 1 | | | | | | Site 2 | | | | | | Site 3 | | | | | | Total | |
|---|---|---|---|---|---|---|---|---|---|---|---|---|---|---|---|---|---|---|---|---|
| | Orchard | | | Surrounding | | | Orchard | | | Surrounding | | | Orchard | | | Surrounding | | | | |
| | No | % | f | No | % | f | No | % | f | No | % | f | No | % | f | No | % | f | No | % |
| *Chrysotoxum vernale* | 3 | 3.7 Sd | 5.6 | | | | 1 | 0.6 Sr | 2.8 | 1 | 0.4 Sr | 2.8 | 1 | 0.7 Sr | 2.8 | 1 | 0.4 Sr | 2.8 | 7 | 0.7 Sr |
| *Dasysyrphus venustus* (Meig.) | | | | | | | | | | | | | | | | 2 | 0.8 Sr | 2.8 | 2 | 0.2 Sr |
| *Didea fasciata* Macq. | | | | | | | | | | | | | | | | 1 | 0.4 Sr | 2.8 | 1 | 0.1 Sr |
| *Didea intermedia* Loew | | | | | | | 1 | 0.6 Sr | 2.8 | | | | | | | | | | 1 | 0.1 Sr |
| *Epistrophe eligans* (Harr.) | | | | | | | | | | 6 | 2.6 Sd | 2.8 | | | | | | | 6 | 0.6 Sr |
| *Epistrophe melanostoma* (Zett.) | | | | 2 | 1.2 R | 5.6 | | | | 31 | 13.4 Eu | 8.3 | 1 | 0.7 Sr | 2.8 | 6 | 2.5 Sd | 2.8 | 40 | 3.9 Sd |
| *Episyrphus balteatus* (Deg.) | 44 | 54.3 Eu | 38.9 | 81 | 49.7 Eu | 36.1 | 39 | 24.7 Eu | 36.1 | 43 | 18.6 Eu | 25 | 88 | 57.9 Eu | 33.3 | 125 | 51.2 Eu | 50 | 420 | 40.8 Eu |
| *Eupeodes corollae* (Fabr.) | 13 | 16.0 Eu | 22.2 | 26 | 16.0 Eu | 22.2 | 29 | 18.4 Eu | 33.3 | 20 | 8.7 | 25 | 27 | 17.8 Eu | 8.3 | 22 | 9.0 | 33.3 | 137 | 13.3 Eu |
| *Eupeodes lapponicus* (Zett.) | 3 | 3.7 Sd | 5.6 | 1 | 0.6 Sr | 2.8 | 12 | 7.6 D | 11.1 | 2 | 0.9 Sr | 5.6 | | | | 1 | 0.4 Sr | 2.8 | 19 | 1.8 R |
| *Eupeodes latifasciatus* (Macq.) | 1 | 1.2 R | 2.8 | 2 | 1.2 R | 2.8 | 8 | 5.1 D | 8.3 | 14 | 6.1 D | 19.4 | 2 | 1.3 R | 5.6 | 11 | 4.5 Sd | 13.8 | 38 | 3.7 Sd |
| *Eupeodes luniger* (Meig.) | 2 | 2.5 Sd | 2.8 | 1 | 0.6 Sr | 2.8 | 9 | 5.7 D | 8.3 | | | | | | | 1 | 0.4 Sr | 2.8 | 13 | 1.3 R |
| *Melangyna lasiophthalma* (Zett.) | | | | 8 | 4.9 Sd | 2.8 | | | | | | | | | | 1 | 0.4 Sr | 2.8 | 9 | 0.9 Sr |
| *Melanostoma mellinum* (L.) | 2 | 2.5 Sd | 5.6 | 6 | 3.7 Sd | 11.1 | 20 | 12.7 Eu | 16.7 | 3 | 1.3 R | | 2 | 1.3 R | 5.6 | 2 | 0.8 Sr | 5.6 | 35 | 3.4 Sd |
| *Parasyrphus annulatus* (Zett.) | | | | 1 | 0.6 Sr | 2.8 | | | | 30 | 13.0 Eu | 2.8 | | | | 12 | 4.9 Sd | 5.6 | 43 | 4.2 Sd |
| *Platycheirus discimanus* Loew | | | | | | | | | | 1 | 0.4 Sr | 2.8 | | | | | | | 1 | 0.1 Sr |

**Table 4.** *Cont.*

| Species | Site 1 | | | | | | Site 2 | | | | | | Site 3 | | | | | | Total | |
|---|---|---|---|---|---|---|---|---|---|---|---|---|---|---|---|---|---|---|---|---|
| | Orchard | | | Surrounding | | | Orchard | | | Surrounding | | | Orchard | | | Surrounding | | | | |
| | No | % | f | No | % | f | No | % | f | No | % | f | No | % | f | No | % | f | No | % |
| *Platycheirus melanopsis* Loew | 1 | 1.2 R | 2.8 | 1 | 0.6 Sr | 2.8 | 1 | 0.6 Sr | 2.8 | | | | | | | 1 | 0.4 Sr | | 4 | 0.4 Sr |
| *Platycheirus scutatus* (Meig.) | | | | 2 | 1.2 R | 5.6 | 3 | 1.9 R | 2.8 | | | | | | | 2 | 0.8 Sr | 5.6 | 7 | 0.7 Sr |
| *Scaeva pyrastri* (L.) | 1 | 1.2 R | 2.8 | | | | 2 | 1.3 R | 5.6 | 2 | 0.9 Sr | 5.6 | | | | 2 | 0.8 Sr | 5.6 | 7 | 0.7 Sr |
| *Scaeva selenitica* (Meig.) | 1 | 1.2 R | 2.8 | 1 | 0.6 Sr | 2.8 | | | | | | | | | | | | | 2 | 0.2 Sr |
| *Sphaerophoria menthastri* (L.) | | | | 1 | 0.6 Sr | 2.8 | | | | | | | | | | 1 | 0.4 Sr | 11.1 | 2 | 0.2 Sr |
| *Sphaerophoria scripta* (L.) | 6 | 7.4 D | 5.6 | 15 | 9.2 D | 27.8 | 17 | 10.8 Eu | 22.2 | 4 | 1.7 R | 8.3 | | | | 8 | 3.3 Sd | 13.9 | 50 | 4.9 Sd |
| *Syrphus ribesii* (L.) | | | | | | | 2 | 1.3 R | 5.6 | 10 | 4.3 Sd | 5.6 | | | | 3 | 1.2 R | 2.8 | 15 | 1.5 R |
| *Syrphus torvus* O.-S. | 1 | 1.2 R | 2.8 | 5 | 3.1 Sd | 11.1 | 6 | 3.8 Sd | 5.6 | 27 | 11.7 Eu | 5.6 | 4 | 2.6 Sd | 5.6 | 24 | 9.8 D | 16.7 | 67 | 6.5 D |
| *Syrphus vitripennis* Meig. | 2 | 2.5 Sd | 5.6 | 10 | 0.6 Sr | 16.7 | 7 | 4.4 Sd | 13.9 | 37 | 16.0 Eu | 8.3 | 27 | 17.8 Eu | 16.7 | 17 | 7.0 D | 16.7 | 100 | 9.7 D |
| *Xanthandrus comtus* (Harr.) | 1 | 1.2 R | 2.8 | | | | 1 | 0.6 Sr | 2.8 | | | | | | | | | | 2 | 0.2 Sr |
| *Xanthogramma pedissequum* Harr. | | | | | | | | | | | | | | | | 1 | 0.4 Sr | 2.8 | 1 | 0.1 Sr |
| Number of specimens | 81 | | | 163 | | | 158 | | | 231 | | | 152 | | | 244 | | | 1029 | |
| Number of species | 13 | | | 16 | | | 16 | | | 15 | | | 8 | | | 21 | | | | |

Eudominants (Eu) > 10%, Dominants (D) 5.1–10%, Subdominants (Sd) 2.1–5%, Recedents (R) 1.1–2%, Subrecedents (Sr) < 1%. No—number, %—percentage, f—frequency.

Differences in the species richness were observed depending on the habitats. The highest indicator of species richness (8.3) was noted in the boundaries of orchard 3 with diverse vegetation, but surprisingly, orchard 3 had the lowest species richness (3.2) (Table 3).

At most sites, the movement of a large number of species of hoverflies from the surroundings to aphid colonies in orchards was observed; however, exceptions to this rule have also been observed—*E. melanostoma* and *P. annulatus*—numerous in the boundaries of orchard 2—were not noted within orchard 2 (Table 4).

The highest similarity was noted between syrphid species reared from aphid colonies in orchard 2 and those collected in surroundings of this orchard (0.5); this suggested that hoverflies, attracted by flowering plants in the vicinity of the orchard, then migrated to the orchard, and subsequently, females laid eggs in aphid colonies. Unexpectedly, the lowest similarity between syrphids in aphid colonies and those collected in semi-natural habitats was noted on site 3 (0.24) (Table 5).

**Table 5.** Similarity of Syrphidae caught in yellow traps and reared from aphid colonies calculated from the Jaccard classic index.

|  | a.c. 1 |  | a.c. 2 |  | a.c.3 |
|---|---|---|---|---|---|
| O 1 | 0.47 | O 2 | 0.41 | O 3 | 0.44 |
| S 1 | 0.39 | S 2 | 0.5 | S 3 | 0.24 |

O—orchard, s—surrounding, a.c.—aphid colonies.

*3.4. Significant Difference and Correlations between the Syrphidae from Different Habitats*

No statistically significant differences were found between the most abundant syrphid species occurring in different habitats (Table 6).

Based on the results, it can be stated that significant positive correlations were found on site 1 between the number of larvae of Syrphidae collected from aphid colonies and hoverfly adults caught in the yellow traps in this orchard (Table 7).

Surprisingly, no significant correlation was found between the syrphids collected in different habitats on the other IPM sites.

No significant correlations of syrphid species were found between the number of species collected in different habitats (Table 7).

**Table 6.** Source of variation and the number of collected syrphids of *E. balteatus*, *E. corollae* and *S. vitripennis* from different habitats: orchard, surroundings and aphid colonies in the years 2011–2013 (one-way ANOVA).

| Habitat | Site 1 | Site 2 | Site 3 |
|---|---|---|---|
|  | *Episyrphus balteatus* (Deg.) | | |
| orchard | 14.67 ± 8.41 ab | 16.00 ± 6.00 ab | 9.33 ± 6.44 ab |
| surroundings | 37.67 ± 9.70 abc | 12.67 ± 4.67 ab | 41.67 ± 14.23 bc |
| aphid colonies | 8.00 ± 0.58 a | 5.00 ± 1.00 a | 6.33 ± 1.33 a |
|  |  | df = 11 F = 6.05 | |
|  |  | $p = 0.000117$ | |
|  | *Eupeodes corollae* | | |
| orchard | 4.33 ± 2.85 ab | 9.67 ± 4.63 ab | 2.33 ± 1.33 ab |
| surroundings | 8.67 ± 6.67 ab | 6.67 ± 2.91 ab | 7.33 ± 1.45 ab |
| aphid colonies | 3.00 ± 0.58 ab | 2.00 ± 0.58 ab | 3.00 ± 0.58 ab |
|  |  | df = 11 F = 2.55 | |
|  |  | $p = 0.026636$ | |
|  | *Syrphus vitripennis* | | |
| orchard | 1.0 ± 0.0 a | 2.33 ± 0.33 a | 2.33 ± 0.88 a |
| surroundings | 3.33 ± 2.33 a | 12.33 ± 8.95 a | 5.67 ± 3.28 a |
| aphid colonies | 2.00 ± 0.58 a | 1.67 ± 0.33 a | 1.00 ± 0.0 a |
|  |  | df = 11 F = 1.76 | |
|  |  | $p = 0.119848$ | |

Means in columns marked with different letters are significantly different from each other (Tukey multiple comparisons test, $p < 0.05$).

**Table 7.** Pearson correlation coefficients (*p*) between the number of adult syrphid specimens and species collected in yellow traps in the orchards (o) and surroundings (s) and syrphid larvae found in aphid colonies (c) in different habitats.

| | | **Syrphid Specimens** | | |
|---|---|---|---|---|
| | | o/s | o/c | s/c |
| Site 1 | r2 | 0.990596 | **0.998353** | 0.981116 |
| | *p* | 0.087376 | **0.036541** | 0.123917 |
| Site 2 | r2 | 0.356919 | 0.568072 | 0.320584 |
| | *p* | 0.592379 | 0.615379 | 0.792242 |
| Site 3 | r2 | 0.197369 | 0.319044 | 0.990232 |
| | *p* | 0.706932 | 0.617876 | 0.089056 |
| | | Syrphid species | | |
| Site1 | r2 | 0.076923 | 0.923077 | 0.0 |
| | *p* | 0.821088 | 0.178912 | 1.0 |
| Site 2 | r2 | 0.428571 | 0.964286 | 0.617347 |
| | *p* | 0.545629 | 0.121038 | 0.424591 |
| Site 3 | r2 | 0.013158 | 1.000000 | 0.013158 |
| | *p* | 0.926814 | | 0.926814 |

Significant *p* values (*p* < 0.05) are highlighted in bold. Orchards/surroundings—o/s; bold is not necessary orchards/colonies—o/c; bold is not necessary surroundings/colonies—s/c bold is not necessary.

## 4. Discussion

The results indicate an influence of the species-rich and well-developed surroundings of orchards on the occurrence of hoverflies. The data highlight the positive effect of diverse wild vegetation on the species richness and abundance of syrphids. These habitats were more attractive than the orchards for hoverflies, which shows a high dependence on flowers as a source of pollen and nectar that is required for reproduction and survival [19,24,25]. In our study, the presence of aphids feeding on plants in the surroundings of orchards (e.g., *Aphis podagrariae* Schrank, *Aphis sambuci* L., *Aphis urticata* Gmelin, *Dysaphis crataegi* Kalt., *Liosomaphis berberidis* Kalt.) also influenced the presence of syrphids there. Aphids were an alternative food for predatory larvae, and the honey dew (kairomone) they secreted additionally attracted female hoverflies. In natural ecosystems, a large number of species occurring as recedents and subrecedents are observed, and such relationships were noted in the vicinity of orchards with rich vegetation. Our data highlighted the positive effect of diverse wild vegetation on the species richness and abundance of hoverflies. In monoculture landscapes (e.g., orchards), groundcover provides poor floral resources so it is highly probable that the richness of flowering plants in the surroundings of orchards represents a source of pollen and nectar as an alternative food source or place for shelter for overwintering for beneficial insects [5,44–46].

Many authors emphasize the role of the surroundings of fields and orchards e.g., forest edges, hedgerows, and flowering plants as factors influencing the syrphid species richness and thus increasing pollination and pest control [29,47]. According to Albert et al. [15], flower strips attracted the beneficial insects, increased their number in the surroundings of the orchards and thus played a significant role in reducing populations of *D. plantaginea*. Piekarska-Boniecka et al. [48] observed a strong positive influence of diverse populations of flowering plants in the surroundings on the occurrence of syrphids and suggested their later migration into orchards.

Syrphid larvae, as specialized predators, were most numerous in aphid colonies in July, a few days after the observed maximum aphid abundance, suggesting that larval diet is also very important in determining population dynamics of aphidophagous species. We found a positive relationship between the aphid colony size and the number of predators, indicating that females select larger colonies of aphids to lay eggs.

The most numerous syrphid species in all habitats (aphid colonies as well as yellow traps in orchards and boundaries) were *E. balteatus*, *E. corollae* and *S. vitripennis*. The dominance of these species in both environments (surroundings and orchards) indicated

the migration of hoverflies from the vicinity of the orchards to their interior, and next the egg laying in aphid colonies, which in turn reduced the population of aphids on apple trees. The studies by other authors confirm the dominance of these species [49,50]. Data on the role of *E. balteatus* in controlling aphid colonies in orchards can be found in previous works [51–54]. The presence of 20 species of predatory syrphids in apple orchards and their boundaries was reported by Trzciński and Piekarska-Boniecka [49], whereas Piekarska-Boniecka et al. [48] noted 38 syrphid species in the orchard and 49 in their surroundings. In both studies and all habitats, the dominants were *E. balteatus* and *E. corollae*. The numerous occurrences of these species and their role in controlling aphid population due to their wide distribution and easy adoption to different conditions was also reported previously [55–57]. The findings from other studies show the differences in species composition of natural enemies occurring in *D. plantaginea* colonies [51,52,58], which might be related to different regions and climates in which the research was conducted.

However, our hypothesis that semi-natural surroundings would increase the number of hoverflies feeding in aphid colonies was not fully supported, and in many cases, we failed to identify the correlation between the syrphids present in the margins and those present in aphid colonies; it could be concluded that the greater number of hoverflies in the surroundings compared to the orchards may reflect further increased rates of predation in aphid colonies.

The stronger effect of orchard margins on aphid control was confirmed by other authors on studies with annual flowering plants sown as a crop surroundings [59,60] compared to the semi-natural habitats in our study. Such differences could occur due to the differentiated designs of experiments in which the surroundings consisted of a mixture of blooming flowers whereas the control surroundings were very poor (e.g., grass, other crops); the age of wild flowers strip and the season were also very important, as such cases consequently led researchers to observe the positive dependence correlation [28,60,61]. In our study, the surroundings consisting of natural weeds, shrubs or trees were not so attractive for syrphids as diverse flowering plants; hence, no correlation was probably confirmed. Nevertheless, it was important to determine the impact of these different semi-natural sites on the occurrence of beneficial insects in orchards.

Studies by many authors [6,15,28] show that in order to ensure effective biological protection with the use of beneficial insects, activities should not be limited only to the edges of orchards, but sowing flowering plants inside the orchard should be considered to facilitate the movement of predators to target pest species. The most suitable habitats for supporting natural enemies are plants with large flowers or inflorescences with un-concealed nectar (e.g., Apiaceae: *Carum carvi* L. *Coriandrum sativum* L. *Daucus carota* L., *Angelica sylvestris* L.) or large patches of flowers e.g., *Crataegus, Prunus, Sambucus, Rubus* and *Salix* that are preferred by parasitic wasps and hoverflies [16,25,62]. Some authors emphasize the necessity of sowing flowering plants at the beginning of the growing season when the first colonies of aphids are formed [6,20,28]. This is very important because aphids reproduce quickly and even small colonies can grow exponentially; thus, pest feeding weakens the plants and inhibits the shoot growth. Furthermore, trees are covered with honey dew, on which sooty mold develops, thereby covering the leaves and limiting the assimilating area.

Although, in many cases, we failed to identify correlations between the syrphids occurring in the apple surroundings and those present in aphid colonies, based on current and previous researches [10,63], it can be concluded that even small clusters of flowering plants in the margins increase the abundance of beneficial insects within an orchard. The presence of trees and shrubs with herbaceous vegetation creates a suitable place for the development of the hoverflies (shelter and food), increasing their biodiversity.

The predatory larvae of Syrphidae found in large numbers in aphid colonies greatly reduced the number of *A. pomi* and *D. plantaginea*, which should be considered along with chemical treatments to control these pests. Among the Syrphidae collected in the surroundings of orchards in yellow Moericke's traps were all kinds of species which

predatory larvae were feeding in aphid colonies, indicating that the Syrphidae attracted by flowering plants growing in semi-natural surroundings of the orchards lay eggs in aphid colonies later so it should be taken into account in the introduction of integrated production in apple orchards.

## 5. Conclusions

The semi-natural surroundings of orchards with flowering plants increased the abundance of hoverflies occurring in orchards. The higher number of hoverflies in surroundings compared to orchards may further reflect higher rates of predation in aphid colonies. Habitats providing flowering trees, shrubs, and herbaceous plants have been shown to be positive for beneficial insects; however, the contribution of each surrounding will depend on its plant composition, abundance in the landscape, and spatial arrangement. The results should help farmers reduce the use of pesticides in orchards with integrated production by leaving semi-natural habitats around the orchards, thus enriching biodiversity of beneficial insects.

**Author Contributions:** Both authors contributed to this work. E.W.-Ż. designed the experiments, analyzed the data and wrote the paper; and E.W. performed the experiments. All authors have read and agreed to the published version of the manuscript.

**Funding:** This research was supported by the Ministry of Science and Higher Education of Poland as a part of research subsidy to the University of Agriculture in Kraków.

**Institutional Review Board Statement:** Not applicable.

**Informed Consent Statement:** Not applicable.

**Data Availability Statement:** The data presented in this study are available on request from the corresponding author.

**Acknowledgments:** The authors would like to thank Phil Murray from the Centre for Agriculture, Royal Agricultural University (England) for the English proofreading of the manuscript.

**Conflicts of Interest:** The authors declare no conflict of interest.

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
