# Peer review of "Surrounding Semi-Natural Vegetation as a Source of Aphidophagous Syrphids (Diptera, Syrphidae) for Aphid Control in Apple Orchards"

_agriculture, doi:10.3390/agriculture13051040_

Round 1

Reviewer 1 Report (New Reviewer)

Comments and Suggestions for Authors

The overall English and grammar of the manuscript is very poor. Consider the rewriting the abstract, introduction and results, in accordance with the grammar and English writing. The other issues are as follows.

Title; the title is not accurate. Consider rewriting. The title must be like a label, so that the reader can easily understand the importance of the study.

Abstract The abstract is poorly written; we suggest rewriting the abstract to include a proper introduction, materials and methods, results, and proper conclusions drawn by this study. The grammar is very poor in the abstract section. Follow the proper abstract guidelines as per scientific writing. There is only materials and methods, followed by a small result. Please rewrite and add a valid abstract.

Line 14 use word “survival” instead of survive.

Introduction

Line 45 remove – from (larvae).

Line 65-76 please make a one paragraph. Also adjust the hypothesis in the start of the paragraph. It is not goof to write the hypothesis in the current form by presenting the new heading in the introduction section.

There is no review of literature is provided in the introduction. The line 65-67 only say that the little research has been done. Improve this section by adding a new paragraph by providing the proper review of literature and scientific research in the past decades.

Materials and methods:

Line 83- 84- Please elaborate or rewrite the sentence “The distance be-83 tween the IPM orchards was 1-1,5 km”.

Line 86-94 add the common names of the species mentioned. It is not good to add only the scientific names.

Line 86 The author has mentioned the “IPM orchard” what does the author want to say about using the word IPM. IPM is an integrated technique use to control the pest. However author didn’t mentioned that what kind of measures has been taken in the orchards for IPP i.e. Chemicals (which chemicals), mechanical control etc???? Also, explain the same issue in the other portions of the Martials and Methods.

Line 108 ‘grass sward’ which grass ???

Line 145-149 use one paragraph. It is not good to write a manuscript in many paragraphs and headings. Also from line 157-166.

Results

The results are very poorly written. Please consider rewriting.

The figures 1, 2 and 3 are poorly presented. These figures has several errors in format and poorly. There is no proper marking that which color exhibits what characteristics?

In all the figures there are two scales. Mean no of aphids and mean no of syrphids. Both in the same graph. How it is possible that one graph has two scales? Reconsider the figures in the whole manuscript.

Table 1 the standard error is very high as presented in the table. For site 1.

 Moreover, table headings are not according to the format. Consider for whole the manuscript.

Table 4. what is Sr, eu etc mentioned in the table. Identify them in table captions.

Conclusions should be set in one paragraph.

The overall English and grammar of the manuscript is very poor. Consider the rewriting the abstract, introduction and results, in accordance with the grammar and English writing. The other issues are as follows.

Author Response

Dear Reviewer and Editors of Agriculture,

First, let us express our gratefulness for the valuable suggestions and comments on the manuscript. We implemented all changes suggested by Reviewers. We apologize for errors, and we have corrected the text as suggested.

Having taken them into consideration, we would like to address the specific points as follows (answers in italics)

Responses to the Reviewer's comments:

  1. The overall English and grammar of the manuscript is very poor. Consider the rewriting the abstract, introduction and results, in accordance with the grammar and English writing. The other issues are as follows.

I am sorry I am not an English expert so English correction has been made by native speaker (entomologist) Professor Phil Murray, Centre for Agriculture, Royal Agricultural University, Cirencester, Gloucestershire, GL7 6JS, England 

  1. Title; the title is not accurate. Consider rewriting. The title must be like a label, so that the reader can easily understand the importance of the study.

Thank you for your valuable suggestions, but the authors believe that title is adequate for the content of manuscript.

  1. Abstract The abstract is poorly written; we suggest rewriting the abstract to include a proper introduction, materials and methods, results, and proper conclusions drawn by this study. The grammar is very poor in the abstract section. Follow the proper abstract guidelines as per scientific writing. There is only materials and methods, followed by a small result. Please rewrite and add a valid abstract.

Thank you very much - the Abstract has been corrected and supplemented as much as possible unfortunately according to the guidelines for authors from Agriculture, the abstract may contain only 200 words, therefore, it is not possible to discuss the methods, results or conclusions in more detail.

  1. Line 14 use word “survival” instead of survive.

It has been corrected.

  1. Introduction

Line 45 remove – from (larvae).

It has been corrected

  1. Line 65-76 please make a one paragraph. Also adjust the hypothesis in the start of the paragraph. It is not goof to write the hypothesis in the current form by presenting the new heading in the introduction section.

Thank you for your valuable advice – it has been corrected.

  1. There is no review of literature is provided in the introduction. The line 65-67 only say that the little research has been done. Improve this section by adding a new paragraph by providing the proper review of literature and scientific research in the past decades.

This section has been improved.

  1. Materials and methods:

Line 83- 84- Please elaborate or rewrite the sentence “The distance be-83 tween the IPM orchards was 1-1,5 km”.

I am sorry I can’t change it because there is no such error in my version of the manuscript.

I suppose these differences may be due to the different  versions of Word program we use.

 I am attaching a print screen of my manuscript

  1. Line 86-94 add the common names of the species mentioned. It is not good to add only the scientific names.

Thank you very much for drawing my attention to such an important detail - the common names have been added.

  1. Line 86 The author has mentioned the “IPM orchard” what does the author want to say about using the word IPM. IPM is an integrated technique use to control the pest. However author didn’t mentioned that what kind of measures has been taken in the orchards for IPP i.e. Chemicals (which chemicals), mechanical control etc???? Also, explain the same issue in the other portions of the Martials and Methods.

Line 82- IPM integrated pest management - is used to specify the method of pest control in apple orchards.

Line 110-112- chapter Material and Methods has been supplemented:

“During the period of the study in the orchards with integrated pest management (IPM) the apple trees were protected using the pesticides and fungicides in accordance with the methodology of integrated production (5-7 treatments against diseases and 6-7 against pests)”.

In all orchards chemical treatments were carried out according to the methodology of integrated protection. According to the authors, there is no need to give the names of chemicals that are different in different countries and the choice of chemicals changes very often.

  1. Line 108 ‘grass sward’ which grass ???

In the examined orchards there was grass(sward) in the inter-rows of trees ;

“grass sward’ - this phrase has been added by native speaker so I have used it in this form

I have added the print screen of English correction

  1. Line 145-149 use one paragraph. It is not good to write a manuscript in many paragraphs and headings. Also from line 157-166.

It has been corrected

  1. Results

The results are very poorly written. Please consider rewriting.

It has been checked

  1. The figures 1, 2 and 3 are poorly presented. These figures has several errors in format and poorly. There is no proper marking that which color exhibits what characteristics?

The graphs show a comparison of the population dynamics of hoverflies and aphids in individual decades in the months of the study. To make the comparison clearly visible, two-scales in one graph were used. On each graph there is a legend describing the colors of the aphid lines  and the hoverfly bars .

Thank you for pointing out that the colors in figures are not clear - colors of lines and bars have been changed to more contrasting ones.

  1. In all the figures there are two scales. Mean no of aphids and mean no of syrphids. Both in the same graph. How it is possible that one graph has two scales? Reconsider the figures in the whole manuscript.

In entomological research, we always use graphs with two scales(special function in Excel) when we want to compare population dynamics of insects numerically very different because it is not possible to present such different values (no of aphids and no of syrphids)on one scale. If we used the same scale for aphids and hoverflies, the bars with hoverflies values would be invisible.

  1. Table 1 the standard error is very high as presented in the table. For site 1.

Thank you for drawing attention to this table- such a high indicator results from the varied number of hoverflies in individual years of the study, it is related, among others, to climatic factors. The data has been recalculated and the results are correct

 Moreover, table headings are not according to the format. Consider for whole the manuscript.

It has been corrected - the differences in appearance are probably related to different versions of Word.

  1. Table 4. what is Sr, eu etc mentioned in the table. Identify them in table captions.

Thank you very much for the suggestion- I missed it-- now I have completed it.

Eudominants (Eu) > 10%, Dominants (D) 5,1-10%, Subdominants (Sd) 2,1-5%, Recedents (R)1,1-2%, Subrecedents (Sr)<1

  1. Conclusions should be set in one paragraph.

It has been corrected.

  1. Comments on the Quality of English Language

The overall English and grammar of the manuscript is very poor. Consider the rewriting the abstract, introduction and results, in accordance with the grammar and English writing. The other issues are as follows.

I am sorry – I am not native speaker so the English correction has been made by native speaker (entomologist) Professor Phil Murray, Centre for Agriculture, Royal Agricultural University, Cirencester, Gloucestershire, GL7 6JS, England. 

Thank you one more for you work and valuable comments and  suggestions which could help us to improve the manuscript.

We hope that the manuscript now meets all the requirements to be published in Agriculture .

                                                                                  Sincerely,

                                                                                  on behalf of co-authors

Elżbieta Wojciechowicz-Żytko

Reviewer 2 Report (New Reviewer)

This manuscript studied the attraction of semi-natural vegetation around apple orchard to aphidophagous syrphids. The results confirmed the important role of surrounding vegetation on the conservation of aphid predators. The quality of the manuscript was average. 

(1) Although I am not a native speaker, I also found some mistakes or grammar errors, eg in line 35, the wider environment, which include shorter life or lower fecundity.

(2) I suggest to present a map of the experimental area.

(3) If possible, provide the population dynamics of aphids in apple orchards with different semi-natural vegetation surroundings.

 The manuscript suffered from some drawbacks in English quality: 

(1) Although I am not a native speaker, I also found some mistakes or grammar errors, eg in line 35, the wider environment, which include shorter life or lower fecundity.

(2) line 40. Aphids are an important biotic stress factor negatively affecting the quality and quantity of the crop. What crop? 

(3) line 41-42: Some aphid species noted on apple trees are the most dangerous pests causing the deformations of leaves and fruits, shoot twisting and chlorosis. I think this sentence is grammatically incorrect.

There should be many similar drawbacks that need to be carefully revised by the authors.

Author Response

Dear Reviewer and Editors of Agriculture,

First, let us express our gratefulness for the valuable suggestions and comments on the manuscript. We implemented all changes suggested by Reviewers. We apologize for errors, and we have corrected the text as suggested.

Having taken them into consideration, we would like to address the specific points as follows (answers in italics)

English correction has been made by native speaker (entomologist): Professor Phil Murray, Centre for Agriculture, Royal Agricultural University, Cirencester, Gloucestershire, GL7 6JS, England 

Responses to the Reviewer's comments:

  • Although I am not a native speaker, I also found some mistakes or grammar errors, eg in line 35, the wider environment, which include shorter life or lower fecundity.

I am sorry- I am not an expert in English so I corrected the text according to the instructions of native speaker- I have attached a print screen of English correction

(2) I suggest to present a map of the experimental area.

Thank you very much for the suggestion but the authors believe that the map does not bring any new information to the methods - the exact geographical coordinates of the research site are given in the chapter.

(3) If possible, provide the population dynamics of aphids in apple orchards with different semi-natural vegetation surroundings.

Each of the graphs (Fig.1-3) shows the population dynamics of aphids and hoverflies in particular apple orchards with different semi-natural vegetation surroundings.

Comments on the Quality of English Language

 The manuscript suffered from some drawbacks in English quality: 

  • Although I am not a native speaker, I also found some mistakes or grammar errors, eg in line 35, the wider environment, which include shorter life or lower fecundity.

I am sorry - I am not an expert of English language therefore the text was checked by native speaker and corrected according to his instructions.

 (2) line 40. Aphids are an important biotic stress factor negatively affecting the quality and quantity of the crop. What crop? 

Thank you very much for the suggestion - it has been corrected.

Aphids are an important biotic stress factor negatively affecting the quality and quantity of the fruit crop [13].

(3)line 41-42: Some aphid species noted on apple trees are the most dangerous pests causing the deformations of leaves and fruits, shoot twisting and chlorosis. I think this sentence is grammatically incorrect.

I am sorry- I am not an expert of English language therefore the text was checked by native speaker and corrected according to his instructions

There should be many similar drawbacks that need to be carefully revised by the authors.

Thank you very much for the valuable suggestion – all text was carefully revised.

Thank you one more for you work and valuable comments and suggestions which could help us to improve the manuscript.

We hope that the manuscript now meets all the requirements to be published in Agriculture .

                                                                                  Sincerely,

                                                                                  on behalf of co-authors

Elżbieta Wojciechowicz-Żytko

Reviewer 3 Report (New Reviewer)

The manuscript entitled “Surrounding Semi-Natural Vegetation as a Source of Aphidophagous Syrphids (Diptera, Syrphidae) for Aphid Control in Apple Orchards” is interesting, giving important information, supporting the significant role of natural vegetation on biological control of insect pests.

I consider that in is suitable for publication in the journal “Agriculture” after minor revision.

Some typing errors are referred below and some remarks that the authors could consider improving their manuscript:

Line 45: “(larvae). The” a line should be omitted

Line 105: instead of “cathartica L;,” please write “cathartica L.,”

In Material and methods, the variety of apple trees could be referred. Also, it would be interesting if any information is available about the dominant species of aphids on the semi-nature vegetation.

Lines 136-137: Give please if it is possible a quantitative information: how mane time were collected the syrphids in each sampling? (about 20min? 1 hour? …) because in table 1 the numbers of collected syrphids are given.

Line 168: Because ethe title is written in italics, the Latin names should not be written in italics. Additionally the size of the letters of “on apple trees” should be corrected.

Line 370: Please correct the “Apiaceae -Carum carvi” to “Apiaceae: Carum carvi”.

Author Response

Dear Reviewer and Editors of Agriculture,

First, let us express our gratefulness for the valuable suggestions and comments on the manuscript. We implemented all changes suggested by Reviewers. We apologize for errors, and we have corrected the text as suggested.

Having taken them into consideration, we would like to address the specific points as follows (answers in italics)

English correction has been made by native speaker:

Professor Phil Murray, Centre for Agriculture, Royal Agricultural University, Cirencester, Gloucestershire, GL7 6JS, England 

Responses to the Reviewer's comments:

  1. Line 45: “(larvae). The” a line should be omitted

Thank you very much for noticing the error - It has been corrected

  1. Line 105: instead of “cathartica L;,” please write “cathartica L.,”

It has been corrected

  1. In Material and methods, the variety of apple trees could be referred. Also, it would be interesting if any information is available about the dominant species of aphids on the semi-nature vegetation.

Thank you very much for this suggestion - cultivars: ‘Szampion’, ‘Elise’ and ‘Elstar’- it was corrected

in three apple orchards with integrated pest management (IPM) with ‘Szampion’, ‘Elise’ and ‘Elstar’cultivars…..

On semi-natural vegetation the occurrence of some  species of aphids was noted : Aphis podagrariae Schrank, Aphis sambuci L., Aphis urticata Gmelin, Dysaphis crataegi  Kalt., Liosomaphis berberidis Kalt, but their numbers and dominance were not studied.

  1. Lines 136-137: Give please if it is possible a quantitative information: how mane time were collected the syrphids in each sampling? (about 20min? 1 hour? …) because in table 1 the numbers of collected syrphids are given.

It was corrected

In order to determine the species composition of Syrphidae occurring in Aphis pomi Deg. and Dysaphis plantaginea Pass  colonies larvae and pupae were collected during 30 minutes from aphid colonies (from the same 10 tress as aphids but from unmarked shoots) from April to July at 10-14 day intervals.

  1. Line 168: Because the title is written in italics, the Latin names should not be written in italics. Additionally the size of the letters of “on apple trees” should be corrected.

I am sorry I can’t change it because there is no such error in my version of the manuscript – I suppose that these differences are due to different versions of Word we use.

I am sending the print screen

  1. Line 370: Please correct the “Apiaceae -Carum carvi” to “Apiaceae: Carum carvi”.

It has been corrected

Thank you one more for you work and valuable comments and  suggestions which could help us to improve the manuscript.

We hope that the manuscript now meets all the requirements to be published in Agriculture .

                                                                                  Sincerely,

                                                                                  on behalf of co-authors

Elżbieta Wojciechowicz-Żytko

Round 2

Reviewer 2 Report (New Reviewer)

The quality of the manuscript has greatly improved after the author's revisions Although the author did not provide a layout map of the experimental site, I still recommend that this version be accepted for publication.

Author Response

Dear Reviewer,

We would like to thank one more for taking Your time, thoughtful, valuable comments  and efforts towards improving our manuscript.

We hope that the manuscript now meets all the requirements to be published in Agriculture .

                                                                                  Sincerely,

                                                                                  on behalf of co-authors

Elżbieta Wojciechowicz-Żytko

This manuscript is a resubmission of an earlier submission. The following is a list of the peer review reports and author responses from that submission.

Round 1

Reviewer 1 Report

Aphids are the main pest insects affecting the production of apple, it is difficult to control, insecticides were usually applied in the control of aphids, but it caused the 3R problem. The manuscript reported the influence of different semi-natural surroundings adjacent to apple orchards,it will provide a ecological method to control aphids.

the ecological control strategy was addressed by the research, the topic was relevant in the field, it did not addressed a specific gap in the field, the references were appropriate. It can be published after minor revisions.

1)     Line 12  delete “obtained in the study”.

2)     Line 122: “10 shoots per tree…”, It is not clear that how to select 10 shoots in a tree, which site on a tree, the top, upper, middle or lower shoot?

3)     Line 124 : “and the aphids colonies were estimated”, all the aphids including alate ahpids and apterous ahpids, or only alate ahpids. Moreover, “the aphids colonies were estimated”, but how to estimate the aphids colonies, number them one by one , or  use the class level,it is not clear.

4)     Line 131:  “….10 traps were placed in each orchard”, it is not clear that how about the methods that 10 traps were placed in each orchard, placed randomly or in a 5 spot site sampling.

5)     Line 132:  “…. a certain distance from…” , how long the distance, should be a certain.

6)     Line 184-185:  “They weakened the plants, inhibited 184 the shoot growth, trees were covered with honey dew on which sooty mold developed 185 covered the leaves and limited the assimilating area.” May be put it in the discussion.

7)     Line 189-190:  “few days later then 189 pests” “few days later after pests”

8)     Line268-269: delete “The Jaccard index [27] was used to compare syrphid communities in aphid colonies and their surroundings (Table 5). ”

9)     Line273: add “,” after the “Unexpectedly”

10)  Line275: Table5 is difficult to review, please change the form of table, or delete the table and describe the results with words。

11)  Line283-284: Delete “ Correlations between hoverflies larvae collected in aphids colonies and adults caught into traps in orchards and their surroundings at each site were calculated (number of species and number of specimens).”

12)  Line286: Delete“ obtained”.

13)  Line286: Delete “obtained in the study”.

14)  Line307: Delete “Our data”.

15)  Discussion: needs to be simplified and be concise. Authors used many short paragrahps in the discussion part, it is difficult to understand for readers. Moreover, authors should compare the results in this study with other studies, especially analyze the difference with the references.

16)Line 79-119  2.1. Research sites: How about the distance between those four test sites.

17)Line121-126 2.2.1. Sampling of aphid: Aphids were inspected and the aphids colonies were estimated, but it is unclear that how to inspect the aphids on marked shoots, umber aphids one by one or using the class level?

Author Response

Dear Reviewer and Editors of Agriculture,

First, let us express our gratefulness for the valuable suggestions and comments on the manuscript.

We implemented all changes suggested by Reviewers. We apologize for errors, and we have corrected the text as suggested.

Having taken them into consideration, we would like to address the specific points as follows (answers in italics)

English correction has been made by native speaker:

Professor Phil Murray, Centre for Agriculture, Royal Agricultural University, Cirencester, Gloucestershire, GL7 6JS, England 

Responses to the reviewer's comments

Reviewer: 1

  Line 12  delete “obtained in the study”.

 Has been deleted

2)     Line 122: “10 shoots per tree…”, It is not clear that how to select 10 shoots in a tree, which site on a tree, the top, upper, middle or lower shoot?

Thank you for valuable suggestion. The chapter “Methods”  has been completed.

To assess the presence of green apple aphid A. pomi and rosy apple aphid D. plantaginea, 100 marked shoots per orchard (10 shoots per tree randomly selected from different parts of the crown in 10 trees, placed up to 20 m away from the margin) were inspected from the end of April to October (at two-week intervals).

3)     Line 124 : “and the aphids colonies were estimated”, all the aphids including alate aphids and apterous aphids, or only alate aphids. Moreover, “the aphids colonies were estimated”, but how to estimate the aphids colonies, number them one by one , or  use the class level,it is not clear.

In small colonies all the aphids including alate and apterous were counted, when they were very numerous their number was estimated.

4)     Line 131:  “….10 traps were placed in each orchard”, it is not clear that how about the methods that 10 traps were placed in each orchard, placed randomly or in a 5 spot site sampling.

Traps were hung on branches 1.5-2m above the ground level, 10 traps were placed randomly in the middle in each orchard and 10 in its boundaries, every 10 m.

5)     Line 132:  “…. a certain distance from…” , how long the distance, should be a certain.

To avoid the impact of marginal effects, the traps were placed 10-15 m from the edge of the orchard or its surroundings.

6)     Line 184-185:  “They weakened the plants, inhibited 184 the shoot growth, trees were covered with honey dew on which sooty mold developed 185 covered the leaves and limited the assimilating area.” May be put it in the discussion.

Has been moved to discussion

7)     Line 189-190:  “few days later then 189 pests” “few days later after pests”

Has been corrected

8)     Line268-269: delete “The Jaccard index [27] was used to compare syrphid communities in aphid colonies and their surroundings (Table 5). ”

Has been deleted

9)     Line273: add “,” after the “Unexpectedly”

I don't know why put the quotation marks - it's not a quoted statement but  it is my observation

 10)  Line275: Table 5 is difficult to review, please change the form of table, or delete the table and describe the results with words。

Has been corrected

11)  Line283-284: Delete “ Correlations between hoverflies larvae collected in aphids colonies and adults caught into traps in orchards and their surroundings at each site were calculated (number of species and number of specimens).”

Has been deleted

12)  Line286: Delete“ obtained”.

Has been deleted

13)  Line286: Delete “obtained in the study”.

Has been deleted

14)  Line307: Delete “Our data”.

Has been deleted

15)  Discussion: needs to be simplified and be concise. Authors used many short paragrahps in the discussion part, it is difficult to understand for readers. Moreover, authors should compare the results in this study with other studies, especially analyze the difference with the references.

Thank you for your valuable suggestion – the discussion has been corrected.

16)Line 79-119  2.1. Research sites: How about the distance between those four test sites.

The distance between the IPM orchards was 1-1,5 km, while the ecological one was 5 km away.

17)Line121-126 2.2.1. Sampling of aphid: Aphids were inspected and the aphids colonies were estimated, but it is unclear that how to inspect the aphids on marked shoots, number aphids one by one or using the class level?

In small colonies all the aphids including alate and apterous were counted, when they were very numerous their number was estimated.

Thank you one more for you work and valuable comments and suggestions which helped us to improve the manuscript.

We hope that the manuscript now meets all the requirements to be published in Agriculture .

                                                                                  Sincerely,

                                                                                  on behalf of co-authors

Elżbieta Wojciechowicz-Żytko

Reviewer 2 Report

Dear authors, here are my comments on your manuscript:

1. Introduction:

- Paragraphs two and three should go together, since they are talking about a similar topic.

-Line 46 and 47: the sentence is not clear, and you should add what vegetation gives to beneficial insects in front of crop: pollen, nectar,...

-The aim is too general and vague, there is not a null hypothesis. The design does not check properly the aim proposed.

2. Materials and methods:

-Since there are three IPM sites and one ecological site, the design is not balanced and you cannot compare the sites.

-Line 126: add identification in point 2.2.2.

-Line 150: explain better what mean those percentages and the dominance classess.

-Line 165: Which statistical software package?

3. Results:

-Figure 1 is confusing. There are no error bars, and you can not mix the three years in the same picture. To compare sites, you should put each aphid in one picture by each year. However, since sites 1, 2 and 3 are similar, I would recommend comparing IPM to ecological. The best approach is three IPM sites and three Ecological sites.

-Table 1: Syrphids on aphid colonies is an average per site or per tree or both? You have data from three years, the mean is the overall data, an average of the three years? Explain, please.

-Table 2: parasitism are not mention in your aim and methodology

-Line 208: mean 29,67 from 89 specimens where it comes from?

-Table 3: You did not explain in methods how you have been assessed richness.

-From line 218 to line 228 is data from table 2 and should be before data from table 3.

-In the text, you have divided syrphids from yellow traps and froma aphid colonies, but you have mixed both in the tables. Reconsider that separation.

-Table 5: in methods, you mentioned that similarity between habitats was checked with an ANOVA. Where are the statistics of the analysis? And Why do you say that ones are more similar than others?

-Table 6: I did not find the ANOVA results, it is really strange that in some specimens and sites there are not differences, even the huge differences like "Syrphus vitripennis" for site 2.

-Table 7: p-values are very high, sometimes 1, and you did not highlight in bold. Explain why.

4. Discussion: You mention a hypothesis that you never state in the introduction. However, you concluded that semi-natural surroundings would not increase overflies in the orchards. I think it's because of the lack of variation in ecological sites in comparison to IPM.

Honestly, I think that your design is not enough robust to get your initial aim. I would recommend simplifying the work and focus in the comparison IPM to ecological.

Author Response

Dear Reviewer and Editors of Agriculture,

First, let us express our gratefulness for the valuable suggestions and comments on the manuscript.

We implemented all changes suggested by Reviewers. We apologize for errors, and we have corrected the text as suggested.

Having taken them into consideration, we would like to address the specific points as follows (answers in italics)

English correction has been made by native speaker:

Professor Phil Murray, Centre for Agriculture, Royal Agricultural University, Cirencester, Gloucestershire, GL7 6JS, England 

Responses to the reviewer's comments

Reviewer: 2

  1. Introduction:

- Paragraphs two and three should go together, since they are talking about a similar topic.

Has been corrected

-Line 46 and 47: the sentence is not clear, and you should add what vegetation gives to beneficial insects in front of crop: pollen, nectar,...

Has been corrected

-The aim is too general and vague, there is not a null hypothesis. The design does not check properly the aim proposed.

 Thank you for the suggestion – the aim has been corrected an hypothesis was added.

The aim of work was to determine species compositions of predatory syrphids occurring in yellow traps in orchards and surroundings and in aphids colonies, determine the influence of the surrounding vegetation of orchards on the appearance of zoophagous species of hoverfly reducing aphid populations on apple trees.

Hypothesis - semi-natural surroundings would increased the number of hoverflies feeding in aphid colonies

  1. Materials and methods:

-Since there are three IPM sites and one ecological site, the design is not balanced and you cannot compare the sites.

You are of course right. Unfortunately, due to the lack of an appropriate number of ecological orchards in this region, we were unable to carry out observations in a larger number of them, hence the disproportions. We hesitated for a long time whether include this data in the publication, at last we decided that results are interesting and show the relationship in the ecological orchard.

-Line 126: add identification in point 2.2.2.

Has been added

-Line 150: explain better what mean those percentages and the dominance classess.

It is the percentage share of individual species of hoverflies. On the basis of the work Kasprzak; NiedbaÅ‚a (1981) there are 5 classes of dominance depend on percentage share. According the authors it was assumed that species whose% is over 10% belong to eudominants 5.1–10% dominants, 2.1–5% subdominants, 1.1–2% recedents, and <1% subrecedents

-Line 165: Which statistical software package?

I am sorry I missed it - Statistica software package

  1. Results:

-Figure 1 is confusing. There are no error bars, and you can not mix the three years in the same picture. To compare sites, you should put each aphid in one picture by each year. However, since sites 1, 2 and 3 are similar, I would recommend comparing IPM to ecological. The best approach is three IPM sites and three Ecological sites.

Thank you very much for valuable suggestion .

I made the new Figure containing all years of observation  and I think it will be difficult to recognize all lines and bars, that's why I suppose that  the first version of the Figure will be easier to read. Although this is an average, it shows the general trend of population dynamics of syrphids and aphids in different sites. 

-Table 1: Syrphids on aphid colonies is an average per site or per tree or both? You have data from three years, the mean is the overall data, an average of the three years? Explain, please.

This is an average of three years per site (syrphids from traps in orchards, traps in surroundings and from aphid colonies, respectively) it was a global comparison of the number of syrphid species and specimens occurring on different sites

-Table 2: parasitism are not mention in your aim and methodology

Of course you are right but during the rearing the  hoverfly larvae,  from syrphid pupae, apart adults, parasitoids also hatched - we thought it was an interesting detail and that's why we added it to the table 2.

-Line 208: mean 29,67 from 89 specimens where it comes from?

You are right – I should explain it- in total during 3 years of observation 89 specimens were collected so per one year average is 29.67

-Table 3: You did not explain in methods how you have been assessed richness.

The species richness was calculated based on formula:

S = s-1/logN

S- species richness

s-total number of different species

N- total number of individuals

-From line 218 to line 228 is data from table 2 and should be before data from table 3.

Has been corrected

-In the text, you have divided syrphids from yellow traps and from aphid colonies, but you have mixed both in the tables. Reconsider that separation.

Table 2 contains data on syrphids collected from aphid colonies, in table 4 there are data on syrphids caught into the yellow traps in orchards and surroundings.

In the other tables there is the comparison of syrphid communities in different habitats therefore they are placed together.

-Table 5: in methods, you mentioned that similarity between habitats was checked with an ANOVA. Where are the statistics of the analysis? And Why do you say that ones are more similar than others?

Similarity of Syrphidae caught into yellow traps and reared from aphids colonies was calculated from Jaccard classic index.

Jclass = A/A + B + C

Jclass- Jaccard similarity index, A- number of shared species, B - number of species unique to the first assemblage, C -number of species unique to the second assemblage.

The result of a mathematical operation is given as the decimals and can be converted to percentages as these are easier to interpret.

The higher the percentage, the more similar  two populations are– I based on this argument.

The differences between the occurrence of the most abundant syrphid species in different habitats were calculated using two-factor Anova statistics-  the data are in table 6

-Table 6: I did not find the ANOVA results, it is really strange that in some specimens and sites there are not differences, even the huge differences like "Syrphus vitripennis" for site 2.

The POST HOC test used for the calculations the data from table 6  is TUKEY's HSD test, which is quite restrictive and therefore the statistical significance of the differences was not obtained.

 If we used POST HOC NIR there would be statistical differences because this test is less restrictive (sensitive).

-Table 7: p-values are very high, sometimes 1, and you did not highlight in bold. Explain why.

The correlation is statistically significant at p<0.05

at p=1, r is not given (not calculated)

at p=1 the correlation is statistically insignificant

  1. Discussion: You mention a hypothesis that you never state in the introduction. However, you concluded that semi-natural surroundings would not increase overflies in the orchards. I think it's because of the lack of variation in ecological sites in comparison to IPM.

Thank  you for the comments -  I have stated the hypothesis.

Our hypothesis was not fully supported probably because we studied only the semi-natural habitats  that were around all the orchards, so we did not obtain such effect and correlations as researchers examining the impact of flowering plant mixtures sown around the orchards and comparing their impact with the control  which was the natural poor vegetation.

Honestly, I think that your design is not enough robust to get your initial aim. I would recommend simplifying the work and focus in the comparison IPM to ecological.

Thank you for valuable suggestion.

I put some information regarding the comparison of IPM with ecological sites to the manuscript, changed the discussion. Unfortunately, I had no possibility to  carry out observations in a larger number of ecological orchards in this region so it is difficult to fully compare this sites.

Thank you one more for you work and valuable comments and suggestions which helped us to improve the manuscript.

We hope that the manuscript now meets all the requirements to be published in Agriculture .

                                                                                  Sincerely,

                                                                                  on behalf of co-authors

Elżbieta Wojciechowicz-Żytko

Round 2

Reviewer 2 Report

Dear authors her my comments:

1. Introduction:

-Paragraphs two and three are still not together.

2. Materials and Methods

-Line 197: Add identification in point 2.2.2 "Sampling and identification of Syrphids"

-Line 235: Statistical description is still not complete. You should explain factors and variables in the two-factor ANOVA.

3. Results

- Figure 1: to me, each bar means syrphids for each year and lines are the average for each aphid. So if you showed an average for aphid, why not an average for syrphids? And if you have averages for aphids and syrphids, you have bars. I can accept that you have 3 IPM sites and 1 ecological site, and you decided to show 4 figures. However, you must show correctly all the data.

-Table 6: I do not understand this table. In a two factor ANOVA, there are two factors. It looks that one factor is location (sites) and the other is place in the orchard. In an ANOVA analysis, you have an F a degree of freedom and a p-value for each factor and the intersection. And I do not find this data, only a post-hoc analysis for each column (site?)

Author Response

Dear Reviewer,

Thank you very much for carefully checking the manuscript again. I am so sorry that I was unable to answer and correct all Your suggestions accurately and I missed some of them.

I hope that now I have corrected properly all errors noticed by the Reviewer and explained all inaccuracies.

  1. Introduction:

-Paragraphs two and three are still not together.

I am sorry – I was sure I have done it.

Now it has been corrected. 

  1. Materials and Methods

-Line 197: Add identification in point 2.2.2 "Sampling and identification of Syrphids"

Syrphid adults collected from yellow traps and those from aphid colonies were identified to species in the laboratory under the microscope taking into account the specific characteristics of each species, based on the keys of  van Veen [32] and Rotheray[33], terminology used was according to SoszyÅ„ski [34].

-Line 235: Statistical description is still not complete. You should explain factors and variables in the two-factor ANOVA.

We are very grateful for noticing the error– you are right- in Methods we by misunderstanding  wrote two-factor ANOVA  but it should be one-way ANOVA- sorry for the mistake! 

  1. Results

- Figure 1: to me, each bar means syrphids for each year and lines are the average for each aphid. So if you showed an average for aphid, why not an average for syrphids? And if you have averages for aphids and syrphids, you have bars. I can accept that you have 3 IPM sites and 1 ecological site, and you decided to show 4 figures. However, you must show correctly all the data.

decades

In Figures 1-4 Population dynamics of syrphids and aphids are shown during the vegetation period – every 10 days (decade )from May to September.

Both:   bars (syrphids) and lines – (aphids)  are three years average .

Each Figure shows one site (1,2,3 are sites with IPM), and site 4 is ecological site.

-Table 6: I do not understand this table. In a two factor ANOVA, there are two factors. It looks that one factor is location (sites) and the other is place in the orchard. In an ANOVA analysis, you have an F a degree of freedom and a p-value for each factor and the intersection. And I do not find this data, only a post-hoc analysis for each column (site?)

I am very sorry but I misunderstood the Reviewer's earlier comment  - I thought it was about the test which was used, so I described the POST HOC test.

We have not been noticed so far to the values of the analysis of variance, therefore they have not been included in the table 6. Now it has been corrected. The Table 6 has been supplemented.

For Syrphus vitripennis, there are no differences between the means because the ANOVA  parameters are not statistically significant p=0.11.

Table 6. Source of variation and the number of  collected syrphids of E.balteatus, E. corollae and S. vitripennis   from different habitats in the years 2011-2013 (one-way ANOVA).

habitat

Site 1

Site 2

Site 3

Site 4

Episyrphus balteatus (Deg.)

orchard

14.67±8.41 ab

16.00±6.00 ab

9.33±6.44 ab

55.67±5.78 c

surroundings

37.67± 9.70 abc

12.67±4.67 ab

41.67±14.23 bc

18.33±2.85 ab

aphid colonies

8.00± 0.58 a

5.00±1.00 a

6.33±1.33 a

14.67±1.76 ab

df=11   F=6.05

   p=0.000117

Eupeodes corollae

orchard

4.33±2.85 ab

9.67±4.63 ab

2.33±1.33 ab

15.67±1.20 b

surroundings

8.67±6.67 ab

6.67±2.91 ab

7.33±1.45 ab

0.00 ab

aphid colonies

3.00±0.58 ab

2.00±0.58 ab

3.00±0.58 ab

8.67±0.88 ab

df=11  F=2.55

  p=0.026636

Syrphus vitripennis

orchard

1.0±0.0 a

2.33±0.33 a

2.33±0.88 a

11.67±1.45 a

surroundings

3.33±2.33 a

12.33±8.95 a

5.67±3.28 a

4.33±1.20 a

aphid colonies

2.00±0.58 a

1.67±0.33 a

1.00±0.0 a

4.33±0.33 a

df=11  F=1.76

   p=0.119848

Means in columns marked with different letters are significantly different from each other (Tukey multiple comparisons test, p < 0.05)

                We appreciate Reviewer work to improve our publication. Thank you once again for Your valuable suggestions. We hope that the manuscript now meets all the requirements to be published in Agriculture .

Sincerely,

Elżbieta Wojciechowicz-Żytko
